# Consistencies are All You Need for Semi-supervised Vision-Language Tracking

## ABSTRACT

Vision-Language Tracking (VLT) requires locating a specific target in video sequences, given a natural language prompt and an initial object box. Despite recent advancements, existing approaches heavily rely on expensive and time-consuming human annotations. To mitigate this limitation, directly generating pseudo labels from raw videos seems to be a straightforward solution; however, it inevitably introduces undesirable noise during the training process. Moreover, we insist that an efficient tracker should excel in tracking the target, regardless of the temporal direction. Building upon these insights, we propose the pioneering semi-supervised learning scheme for VLT task, representing a crucial step towards reducing the dependency on high-quality yet costly labeled data. Specifically, drawing inspiration from the natural attributes of a video (i.e., space, time, and semantics), our approach progressively leverages inherent consistencies from these aspects: (1) Spatially, each frame and any object cropped from it naturally form an image-bbox (bounding box) pair for self-training; (2) Temporally, bidirectional tracking trajectories should exhibit minimal differences; (3) Semantically, the correlation between visual and textual features is expected to remain consistent. Furthermore, the framework is validated with a simple yet effective tracker we devised, named ATTracker (Asymmetrical Transformer Tracker). It modifies the self-attention operation in an asymmetrical way, striving to enhance target-related features while suppressing noise. Extensive experiments confirm that our ATTracker serves as a robust baseline, outperforming fully supervised base trackers. By unveiling the potential of learning with limited annotations, this study aims to attract attention and pave the way for Semi-supervised Vision-Language Tracking (SS-VLT).

## CCS CONCEPTS

• **Computing methodologies → Tracking**.

## KEYWORDS

Vision-Language tracking, Semi-supervised learning

## 1 INTRODUCTION

Vision-Language Tracking (VLT), a fundamental and challenging task in multi-modal video understanding, has drawn increasing attention due to its vast applications in intelligent surveillance [40], autonomous driving [37], human-computer interaction [19] and other fields. The primary objective of this task is to localize *target* within video frames (termed *search images*), guided by semantic provided by the language prompt and visual cues dubbed *template*.

To date, all previously proposed methods for VLT are trained under a fully-supervised setting. Despite the remarkable success achieved by these deep learning-based trackers, their data-hungry nature demands a substantial amount of fully annotated data for effective training. Moreover, VLT distinguishes itself significantly from other tasks (e.g., object detection, semantic segmentation),

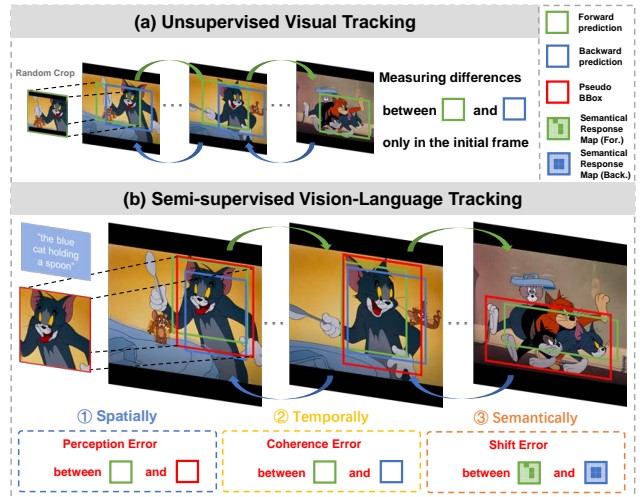

Figure 1: The comparison between the learning scheme of unsupervised visual tracking and our framework for Vision-Language Tracking. The former treats the region randomly cropped from the frame as the target, which is unlikely to contain an object with meaningful context. It then utilizes the prediction deviations of bi-directional tracking in the first frame as supervision signals for training. In contrast, our approach identifies the object with the most relevant semantic to the language prompt as the pseudo target. By further leveraging multi-consistences errors during tracking in raw videos, we train the tracker without heavyweight annotations.

as categories of the target are not restricted to a pre-defined set, naturally introducing an open-world setting [2, 17]. Hence, training a robust tracker requires more than one large dataset to enhance performance in open-world scenarios. For instance, the widely used TNL2k[53] dataset comprises 1,240,000 video frames along with corresponding bounding boxes, and 2,000 matched natural language descriptions. The manual annotation of such vast amounts of data is time-consuming and labor-intensive.

To tap into the wealth of unlabeled videos available on the Internet, several attempts have been made to explore visual tracking without direct supervision [50, 58]. Generally, these efforts strive to extract self-supervision signals from the temporal dimension of videos (shown in Fig. 1(a)). They assume that a robust tracker can accurately predict both forward and backward tracking, even when the target is a randomly cropped region from the search image. Consequently, differences between the bi-directional predictions of the target in the initial frame can serve as a source of supervision.

However, we argue that such a learning framework is unsuitable for training a VL tracker. In contrast to vision-only tracking, the

crux of VLT lies in the integration of an additional natural language prompt. This linguistic input contributes a more precise expression to the target by incorporating high-level semantic information (e.g., attributes, categories, and structural relationships with other objects). Yet, these advantages are valid only when the visual information and linguistic prompt are properly aligned and fused [13]. A randomly selected region in the search image is unlikely to contain an object with meaning, let alone provide multi-modal semantics to facilitate VL representation learning for tracking. On the other hand, random samples lack clear object edges, hindering the regression learning for coordinates prediction. Moreover, unsupervised methods solely rely on visual cues and treat tracking as a template-matching problem in each frame. Without an understanding of the language instruction, these methods struggle in situations such as the ambiguity of bounding box (bbox) and the extensive appearance variation of the target. Therefore, challenges persist in discovering supervision signals from unlabeled data that are beneficial for cross-modal learning in VLT.

In this paper, we concentrate on training a Vision-Language tracker with minimal manual intervention, aiming to bridge the research gap between fully-supervised and unsupervised tracking. We insist that an efficient VL tracker should first understand what to track, and then follow closely to the target regardless of its changes or the direction of the time. To achieve this, the tracker must perceive the semantic from both modalities and align multi-modal information to estimate trajectories in subsequent frames, overcoming challenges such as occlusions and appearance variations. Therefore, we propose a learning scheme (shown in Fig. 1(b)) for VLT that generates high-quality pseudo labels and explores multiple consistencies inherent in raw videos, achieving competitive performances with limited annotations. Specifically,

(1) To equip the tracker with the ability to perceive multi-modal semantic and accurately locate the target, we leverage spatial consistency in each frame, since an image and any object cropped within it naturally form an image-bbox pair for self-training. To ensure the cropped object contains rich semantic and is highly related to the linguistic description, we fine-tune a Vision-Language Pre-trained Model (VLPM) to discover an object with clear edges as pseudo label. Ideally, the identified object is the target of interest. Hence, valuable Vision-Language knowledge are transferred from VLPM via the generated image-bbox-text triplets.

(2) Leveraging the reversibility inherent in the video, we exploit temporal consistency by conducting tracking in both forward and backward directions. Given the initial state of the target, we first track it forwardly across frames. Subsequently, we treat the predicted object in the last frame as the new target and proceed to trace it backward to the initial frame. Ideally, predictions should be identical regardless of the temporal direction. Hence, the differences between the forward and backward trajectories serve as valuable supervision signals.

(3) From a semantic standpoint, the correlation of visual and linguistic features represents the highlights of the tracker's attention, which should remain consistent in bidirectional tracking, mainly focusing on the target.

We integrate the consistencies-based learning framework into a baseline method we devised, named Asymmetrical Transformer Tracker (ATTracker). ATTracker is a simple yet efficient real-time tracker, only consisting of an Asymmetrical Multi-source Encoder (AME) and an MLP-based Decoder. It first tokenizes Multi-source information (i.e., template, search, and language prompt) and sends them to the Asymmetrical Multi-source Encoder, which modifies the conventional Multi-Head Attention [48] in an asymmetrical way. Through this adjustment, AME reinforces the representation of target-related features while effectively suppressing irrelevant background information. Additionally, we introduce a special learnable prediction token, denoted as [REG], and concatenate it with the original tokens from the template, search, and language prompt. It aims to capture the complex relation between multi-modal features for subsequent regression in MLP-based Decoder, functioning similarly to the [CLS] token in standard ViT. Extensive experimental results indicate that without bells and whistles, the proposed semi-supervised tracker achieves comparable performance with the baseline fully supervised VL trackers [31, 53].

To sum up, our main contributions are as follows:

- To the best of our knowledge, we propose the first semi-supervised learning scheme for Vision-Language Tracking, which provides robust supervision signals via mining consistencies inherent in unlabeled videos.
- We devise a simple yet efficient Asymmetrical Transformer Tracker to verify the effectiveness of our framework, serving as a strong baseline for SS-VLT.
- Extensive experiments demonstrate the favorable performance of the proposed method and highlight the potential of learning a VL tracker with limited annotations.

## 2  RELATED WORK

### 2.1  Unsupervised Visual Tracking

The pursuit of training a tracker without annotated data has been a longstanding objective since the inception of this task. As a video-centric task, exploring the temporal reversibility and leveraging the consistencies in bidirectional trajectories become a natural avenue for obtaining supervision signals. Kalal et al. [18] explicitly decompose the long-term tracking task into sub-tasks, and then detect tracking failures via forward-backward matching. Lee et al. [24] analyze geometric consistencies between a pair of forward-backward trajectories to measure the robustness of a base tracker. To harness the capabilities of deep learning, Wang et al. [50] revisit this scheme and train UDT, a Discriminative Correlation Filters (DCF)-based tracker, with the supervision of consistency loss. Subsequently, Zheng et al. [58] propose an unsupervised training approach by initially training from a single frame in the first stage and then adopting cycle training to learn temporal correspondences. Similar ideas can also be found in data annotation and tracker evaluation for TrackingNet [39]. In summary, the success achieved by the aforementioned works confirms the effectiveness of utilizing bidirectional tracking to train robust trackers. However, these approaches merely utilize visual cues and treat tracking as a template-search matching task, relying heavily on the localization ability brought by the correlation filters or CNNs within. Without

understanding the semantic of the target, their performances degrade significantly when encountering vast appearance variations of the target. Different from these works, our method focuses on multi-modal perception with limited labels, leveraging multiple consistencies inherent in videos to achieve superior performance in long-term tracking.

## 2.2 Supervised Vision-Language Tracking

The integration of vision and language presents promising opportunities for enhancing the robustness and addressing inherent limitations in vision tasks [42], including tracking [10, 55]. Compared to initializing a tracker only with a bounding box (bbox), employing a language prompt emerges as a more user-friendly and straightforward mode of communication for humans. This approach not only allows for a precise description of the target but also provides comprehensive semantic details such as attributes and category. With the aid of such abundance information, the tracker can effectively tackle the inherent ambiguity caused by bbox and properly handle the extensive appearance variations of the target. As a pioneering work, Li et al. [31] define the task of Vision-Language Tracking and lay the foundation for subsequent researches. Ma et al. [38] introduce the CapsuleTNL, which utilizes the Capsule Network and two interaction routing modules to effectively integrate image and text features. Wang et al. [53] introduce TNL2K, a new large-scale benchmark for VLT, which stands out due to several desirable features: high quality, adversarial samples and significant appearance variation. SNLT [11] improve tracking performance with their Dynamic Aggregation Module to combine both modalities. JointNLT [60] propose a novel framework that reformulates grounding and tracking as a unified task, focusing on the alignment of multi-modal semantic. Nonetheless, these works rely on separate visual and textual encoders to extract features, without direct interactions between modalities during representation learning. In contrast to these approaches, Guo et al. [13] introduce ModaMixer, a novel module aiming to learn a unified visual-language representation for tracking. SATracker [12] further validates the importance of unified feature extraction and interaction. Despite these advancements, all of them require a large amount of human-annotated data for training. Consequently, we propose a pioneering semi-supervised method for the VLT task, aiming to mitigate the dependency on high-quality yet costly labeled data.

## 2.3 Vision-Language Pre-Trained Models

Vision-Language Pre-Trained Models (VLPMs) have established a dominant presence across vast multi-modal tasks [6, 15, 25, 26], e.g., visual question answering, cross-modal reasoning, and visual grounding. These VLPMs can be generally divided into two categories according to their training frameworks: coarse-grained and fine-grained. Coarse-grained approaches concentrate on extracting and encoding overall image features with convolutional network [16] or vision transformer [20, 35]. While effective, these methods may not perform as well as fine-grained approaches on object-level tasks due to the lack of region-centric features. In contrast, the fine-grained methods [6, 29, 36, 46] utilize a pre-trained object detector [1, 44] as the image encoder. Trained on annotations of common objects [22, 32], these detectors can identify regions

likely to contain objects and perform object classification on each region. However, these methods can only recognize objects within the given categories, such as the 80 object categories in the COCO dataset. There are also some methods [34, 54, 56] attempting to learn both object-level and image-level alignments. In this work, we choose to fine-tune $X^2$-VLM [56] for generating pseudo-labels of objects in each frame based on the language prompt. Notably, $X^2$-VLM distinguishes itself by not relying on a closed-set object detector and learning vision-language alignments in a unified manner, which caters well to our demands of obtaining supervision signals beneficial for multi-modal perception in VLT.

## 3 THE PROPOSED METHOD

### 3.1 Overview

As illustrated in Fig. 2, our semi-supervised learning scheme framework can be generally divided into two stages: data generation and consistences-based training. The first stage in section 3.2 aims to discover the object with semantic that is highly related to the given language prompt. To accomplish this, we fine-tune a Vision-Language Pre-trained Model (VLPM) $X^2$-VLM to visually ground the desire object in each frame, effectively transferring the off-the-shelf Vision-Language knowledge from VLPM to our tracker. The details of the training framework in the second stage are presented in section 3.3, where multi-consistencies from raw videos are exploited to serve as supervision signals. Finally, the baseline Asymmetrical Transformer Tracker (ATTracker) and its learning objective are introduced in section 3.4.

Notably, the **semi-supervised** training setting [47] in Vision-Language Tracking falls conceptually between supervised and unsupervised learning. This setting allows the utilization of large amounts of unlabeled data, commonly available in many scenarios, along with typically smaller sets of labeled data. We adopt the partition setting employed in similar tasks [5, 49], where the proportion of labeled data remains below 50%. In addition, we introduce a novel partition setting of labeled data for VLT, considering the initialization of the VL tracker. In this setting, the tracker can only leverage the **initial information**—comprising visual cues and the language prompt in the first frame—apart from the raw video data.

### 3.2 VLPM-driven Data Generation

When the precise location of the target in each frame is unavailable, the most straightforward and efficient approach for training a Vision-Language (VL) tracker is to generate pseudo bounding boxes that, if possible, closely align with the ground truth [23]. Unlike unsupervised tracking methods that treat randomly cropped regions as pseudo labels [50, 58], we propose to discover the object of interest in each frame of unlabeled videos. To achieve this, we employ $X^2$-VLM [56] to identify the object most relevant to the given language prompt. It is an open-source Vision-Language Pre-trained Model that is designed to image-level and region-level vision-language alignments in a unified manner. Ideally, it can directly predict the location of the target. $X^2$-VLM mainly consists of a visual encoder built on BEiT2 [41], a textual encoder adopted from BERT [8], and a multi-modal fusion module. Trained on vast amounts of datasets and pre-tasks, $X^2$-VLM possesses abundant VL knowledge that can be harnessed for pseudo-label generation.

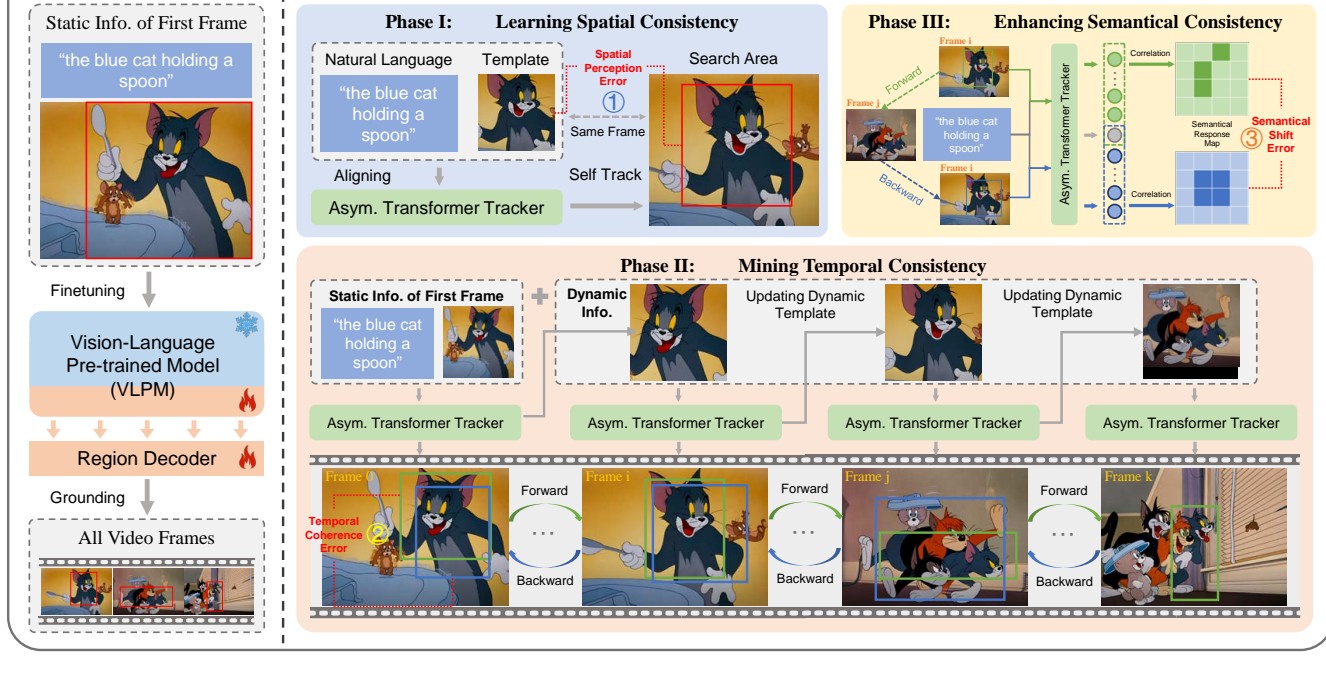

**Figure 2: Overview of the proposed semi-supervised tracking framework. Left: The pseudo label generation pipeline. Right: The detailed structure of the consistencies-based training framework. The ATTracker is first trained to learn spatial consistency from the same frame. Then, we utilize the supervision signals brought by bidirectional tracking to mine temporal consistency and enhance semantical correlation.**

The detailed pipeline of VLPM-driven data generation is illustrated in the left part in Fig. 2. During fine-tuning, we leverage the static information for initializing the VL tracker, i.e., the bbox $B_0^{gt}$ of the template $T_0$, the search image $S_0$, and the language prompt $L$ in the first frame. To prevent high-quality vision-language knowledge of $X^2$-VLM from being disturbed by limited data, we only fine-tune the fusion module for task adaptation while keeping the encoders frozen. Subsequently, the output [CLS] token $X_{cls}$ from $X^2$-VLM is processd by an MLP to predict the bbox $\hat{B}_0$ of the object. For fine-tuning the entire model, we use a linear combination of the classic $L_1$ loss and the scale-invariant Generalized IoU loss [28]:

$$\hat{B}_0 = Sigmoid(MLP(X_{cls}),$$
$$\mathcal{L}_f = \mathcal{L}_{GIoU}(\hat{B}_0, B_0^{gt}) + \mathcal{L}_1(\hat{B}_0, B_0^{gt}). \tag{1}$$

After fine-tuning, the model predicts the bbox $B^{pgt}$ of the most relevant object as pseudo target $T^p$ frame by frame, according to the language prompt of the video. Therefore, we are able to effortlessly collect a substantial quantity of noisy yet valuable image-bbox-text triplets that is beneficial for the subsequent training of the VL tracker.

## 3.3 Progressive Training Framework based on Multiple Consistencies

Despite the abundance of pseudo labels generated in section 3.2, the presence of noise within these training triplets cannot be ignored.

Therefore, directly employing the normal training procedure used by fully-supervised trackers is harmful. To mitigate the negative influence brought by noisy labels, we introduce a training framework that progressively exploit multiple consistencies inherent in raw videos. As depicted in Fig. 2(b), our framework primarily consists of three phases, each focusing on different aspects based on the inherent attributes of a video: space, time, and semantics. By conducting progressive training under this framework, the differences measured in each phase will altogether formulate a multiple consistencies loss. Optimized by this loss, the asymmetrical transformer tracker can make robust predictions of the target's location with limited ground-truth annotations.

*3.3.1 Learning Spatial Consistency.* Inspired by the strategy of curriculum learning [52], we decouple the tracking task into subtasks from easy to hard: image-level perception and video-level trajectory prediction. To grasp image-level semantics, the tracker must align multi-modal information to understand what to track in the first place, then regresses the coordinates of the desired object in each frame based on the multi-modal instructions. In an ideal scenario where the target remains stable, exhibiting minimal appearance variations and no occlusions from the initial state, a VL tracker should be able to locate the target without relying on previous contextual information.

To extract supervision signals from each image, we propose leveraging spatial consistency by instructing the VL tracker to self-track

in the current frame $S_i$. This self-supervised task is based on the observation that an image and any of its sub-regions naturally form a training pair for visual tracking [45]. Specifically, the VL tracker *ATTracker* (illustrated in section 3.4) can only use the language prompt $L_i$ and the pseudo target $T_i^p$ in frame $i$ to perform tracking. Subsequently, the differences between the prediction $\hat{B}_i$ and the bbox $B_i^{pgt}$ of pseudo target $T_i^p$ can be measured:

$$\hat{B}_i = ATTracker(T_i^p, S_i, L_i),$$
$$\mathcal{L}_{space} = \lambda_{giou}\mathcal{L}_{GIoU}(\hat{B}_i, B_i^{pgt}) + \lambda_{L_1}\mathcal{L}_1(\hat{B}_i, B_i^{pgt}). \quad (2)$$

Optimized with a loss function emphasizing spatial consistency during Phase I training, the ATTracker acquires the fundamental ability to align and perceive information from both visual and textual modalities for Vision-Language Tracking.

### 3.3.2 Mining Temporal Consistency.

To enable video-level trajectory predictions, a robust VL tracker needs to address challenges like large motion and appearance variations. Relying solely on spatial consistency for training is insufficient, as it may result in failures to track objects over long temporal spans or in complex scenes. Therefore, we propose to further train the ATTracker by leveraging temporal consistency inherent in the video.

The overall process of Phase II training is illustrated in Fig. 2(b). Given that the direction of time flow in a video is reversible, a robust tracker should be capable of tracking the target efficiently in both forward and backward directions. Therefore, supervision signals are derived from the discrepancies between the trajectories obtained from forward and backward tracking. Specifically, we initiate forward tracking from a template $T_0$ in the initial frame up to frame $k$, with a frame interval of $d$. At each tracking step, we simultaneously crop the predicted object as an online target $T^o$ and concatenate it with the template $T_0$ to capture long-term variations of the target. Here, we illustrate the forward tracking process in frame $i$ as an example:

$$\hat{B}_i^f = ATTracker([T_0; T_{i-d}^o], S_i, L_i), d \leq i \leq k. \quad (3)$$

Then the predicted bbox of forward tracking in frame $k$ are regarded as the new target $T_k$. Likewise, we further conduct backward tracking to the original search area $S_0$ in the first frame:

$$\hat{B}_i^b = ATTracker([T_k; T_{i+d}^o], S_i, L_i), 0 \leq i \leq k - d. \quad (4)$$

Finally, the distinctions between the trajectories obtained from forward and backward tracking in each frame are accumulated, forming the loss function across the temporal dimension:

$$\mathcal{L}_{time} = \sum_{i=0}^{k} \lambda_{giou}\mathcal{L}_{GIoU}(\hat{B}_i^f, B_i^b) + \lambda_{L_1}\mathcal{L}_1(\hat{B}_i^f, B_i^b). \quad (5)$$

By incorporating temporal consistency and the simple online updating strategy into the training process, the ATTracker can effectively follow the target across multiple frames in both directions, thereby enhancing its ability to handle challenges brought by long-term tracking.

### 3.3.3 Enhancing Semantical Consistency.

Recent studies [33, 43, 59] have revealed that the correlation between visual and linguistic features plays a crucial role in directing attention within their models and improving the performance of dense prediction tasks. Motivated by this, we believe that the region of the target should also be strongly correlated with the textual description in VLT. Moreover, such correlation should remain consistent in bidirectional tracking. Thus, we compute the semantical response map using the search feature $f^S$ and the language prompt feature $f^L$ in the final layer:

$$map = upsampling(f^{S'}f^{L'^{\top}}). \quad (6)$$

Here $f^{S'}$ and $f^{L'}$ are L2 Normalized along the channel, $map \in R^{H_{up} \times W_{up}}$, $H_{up}$ and $W_{up}$ are hyperparameters.

To prevent a significant shift of the attention in backward tracking, we treat the response map $map_0^f$ in the initial frame as the pseudo ground truth and compute the consistency of semantic with a binary cross-entropy (BCE) loss:

$$\mathcal{L}_{semantic} = BCE(Sigmoid(map_0^f/\tau), map_0^b/\tau)). \quad (7)$$

Here $\tau$ represents the temperature coefficient.

## 3.4 Asymmetrical Transformer Tracker

Previous unsupervised trackers are constructed either with discriminative correlation filters (DCFs) or CNNs, where the matching operation between the template and search areas plays a critical role in fusing visual features. However, the linear nature of this matching operation often limits the tracker's ability to capture the complex nonlinear interactions among features [4]. Moreover, advancements with the transformer architecture has demonstrated its remarkable efficiency in fusing multi-modal representations, particularly for integrating information from both vision and language modalities [42]. Therefore, we propose the first fully transformer tacker for semi-supervised vision-language tracking (SS-VLT), named Asymmetrical Transformer Tracker (ATTracker). The overall architecture is depicted in Fig. 3. ATTracker takes four types of information as input: static and dynamic templates, search area, natural language, and the learnable prediction token [REG]. The static template represent the state of the target in the initial frame while the dynamic one is periodically updated online at fixed intervals.

Its Asymmetrical Multi-source Encoder (AME) primarily comprises $N$ layers of asymmetrical attention modules (AAMs), where template tokens are restricted to self-attention ($SA$) to mitigate noise disruption. Meanwhile, it facilitates the interaction of other tokens for multi-modal feature learning through asymmetrical cross-attention ($CA$). To elaborate, we begin by applying a linear projection to generate *queries, keys,* and *values* following the standard transformer procedure, which are subsequently concatenated for the asymmetrical attention operation. Here we denote the static template as $Q_{t^s}, K_{t^s}, V_{t^s}$, the dynamic online template as $Q_{t^o}, K_{t^o}, V_{t^o}$, search areas as $Q_s, K_s, V_s$, the language prompt as $Q_l, K_l, V_l$, and the [REG] token as $Q_r, K_r, V_r$. The *queries* are concatenated as follows: $Q_t = \text{Concat}(Q_{t^s}, Q_{t^o})$, $Q_{others} = \text{Concat}(Q_s, Q_l, Q_r)$, and $Q_{all} = \text{Concat}(Q_t, Q_{others})$. Same for the formulation of *keys* and *values* ($K_t, K_{others}, K_{all}, V_t, V_{others}, V_{all}$). In each layer of AAM:

**Figure 3: ATTracker is a fully transformer tracking framework, composed of a transformer backbone and one simple MLP head on the learnable region token. The Asymmetrical Attention Module within it restricts template tokens to performing self-attention to prevent noise. Meanwhile, it facilitates the interaction of other tokens for multi-modal feature learning.**

$$Atten_t = Softmax\left(\frac{Q_t K_t^T}{\sqrt{C}}\right) V_t$$

$$= SA_t V_t,$$

$$Atten_{others} = Softmax\left(\frac{Q_{others} K_{all}^T}{\sqrt{C}}\right) V_c$$

$$= Concat(SA_{others}, CA_{others}) V_{all}, \quad (8)$$

where

$$SA_{others} = Softmax\left(\frac{Q_{others} K_{others}^T}{\sqrt{C}}\right),$$

$$CA_{others} = Softmax\left(\frac{Q_{others} K_t^T}{\sqrt{C}}\right). \quad (9)$$

Here $C$ denotes the dimension of the $key$, and $Atten_t$ and $Atten_{others}$ represent the attention maps for the targets and others, respectively. The attention maps are then passed through a linear layer and added to their respective original tokens using a residual connection.

Afterwards, similar to the [CLS] token used in standard ViT, the outputted [REG] token is sent to a simple MLP to decode the box coordinates. Additionally, the visual and linguistic features of the last layer in AME are leveraged in learning semantical consistency.

Overall, the ATTracker is trained progressively with the combination of multiple consistencies-based losses:

$$\mathcal{L}_{total} = \mathcal{L}_{space} + \mathcal{L}_{time} + \lambda_s \mathcal{L}_{semantic}, \quad (10)$$

where $\lambda_s$ is a hyperparameter.

## 4 EXPERIMENTS

### 4.1 Experiment Setup

**Implementation Details.** The proposed semi-supervised approach is implemented using PyTorch 1.9, running on two NVIDIA RTX 3090 GPUs. We only use the training split of TNL2K [53], OTB99

[31], and LaSOT [9] datasets through the experiments. For optimization, we use ADAM [21] with weight decay of 0.1. Initially, we train the ATTracker following the training procedure outlined in Phase I, which spans 200 epochs. The learning rate for the backbone encoder is set to $4 \times 10^{-5}$, while for the MLP-based decoder, it is $4 \times 10^{-4}$. The learning rates are reduced to one-tenth after the 160th epoch. Subsequently, we fine-tune the model during Phases II and III for an additional 100 epochs each. The learning rate for the backbone encoder is adjusted to $2 \times 10^{-6}$, and for the MLP-based decoder, it is $2 \times 10^{-5}$. Similarly, the learning rates decrease to one-tenth after the 60th epoch. For hyperparameters, the batch size is set to 64 during Phase I and reduced to 16 for Phases II and III. The $\lambda_{giou}$, $\lambda_{L_1}$ and $\lambda_S$ are set to 2,5,0.1. The length of search frames is 4 with an interval of 10. in Phase II. The upsample size of the semantical response map is $40 \times 40$. We set the temperature parameter $\tau = 0.07$.

**Datasets and metrics.** We assess the effectiveness of our approach on mainstream benchmarks specifically tailored for Vision-Language (VL) tracking: TNL2K and OTB99. Additionally, we evaluate our method on LaSOT, a long-term visual tracking benchmark that includes natural language descriptions. These datasets employ success (SUC), precision (PRE), and normalized precision (Norm. PRE) metrics to measure tracking performance.

### 4.2 Comparison with the State-of-the-art Trackers

We conduct a thorough comparison of our semi-supervised learning approach with baseline methods and state-of-the-art (SOTA) trackers under various initialization and supervision settings, as summarized in Table 1.

**TNL2K.** TNL2K, a dataset comprising 2,000 video sequences, presents significant challenges for VLT due to its high-quality content, presence of adversarial samples, and substantial appearance variations. As shown in Table 1, we evaluate the proposed ATTracker with SOTA trackers including JointNLT [60], CTRNLT [30], SNLT [11], TNL2K-2 [53], AutoMatch [57], USOT [58], and LUDT [51].

Table 1: Success (SUC), Precision (PRE), and Normalized Precision (Norm.PRE) of different trackers on the TNL2K, OTB99, and LaSOT. The best and second-best results are marked in bold and underline accordingly. BBox and NL represent the Bounding Box and Natural Language, respectively. We add the symbol * over baseline methods for fair comparisons with our base tracker.

| Algorithms | Published | Supervised | Initialize | TNL2K | | | OTB99 | | LaSOT | |
|---|---|---|---|---|---|---|---|---|---|---|
| | | | | SUC | Norm.PRE | PRE | SUC | PRE | SUC | PRE |
| SiamFC [3]* | CVPR16 | Yes | BBox | 29.0 | 35.0 | 30.0 | — | | 40.0 | 34.0 |
| GradNet [27] | CVPR19 | Yes | BBox | 32.0 | 40.0 | 32.0 | — | | 35.0 | 37.0 |
| AutoMatch [57] | ICCV21 | Yes | BBox | 47.2 | - | 43.5 | — | | 58.3 | 59.9 |
| TNLS-III [31]* | CVPR17 | Yes | NL+BBox | | —— | | 55.0 | 72.0 | — | |
| RTTNLD [10] | WACV20 | Yes | NL+BBox | | —— | | 61.0 | 79.0 | 35.0 | 35.0 |
| TNL2K-2 [53]* | CVPR21 | Yes | NL+BBox | 42.0 | 50.0 | 42.0 | 68.0 | 88.0 | 51.0 | 55.0 |
| SNLT [11] | CVPR21 | Yes | NL+BBox | 27.6 | - | 41.9 | 66.6 | 80.4 | 54.0 | 57.6 |
| CTRNLT [30] | CVPRW22 | Yes | NL+BBox | 44.0 | 52.0 | 45.0 | 53.0 | 72.0 | 52.0 | 51.0 |
| JointNLT [60] | CVPR23 | Yes | NL+BBox | 56.9 | 73.6 | 58.1 | 65.3 | 85.6 | 60.4 | 63.6 |
| **ATTracker*** | **Ours** | Yes | NL+BBox | **56.9** | **75.0** | **64.7** | **69.3** | **90.3** | **63.7** | **67.3** |
| KCF [14] | T-PAMI15 | No | BBox | | —— | | — | | 17.8 | 16.6 |
| ECO [7] | CVPR17 | No | BBox | | —— | | — | | 32.4 | 30.1 |
| UDT [50] | CVPR18 | No | BBox | 27.0 | 37.0 | 31.4 | 59.4 | 76.0 | - | - |
| LUDT [51] | IJCV21 | No | BBox | - | - | - | 60.2 | 76.9 | 26.2 | 23.4 |
| USOT [58] | ICCV21 | No | BBox | 30.0 | 44.1 | 35.7 | 58.9 | 80.6 | 33.7 | 32.5 |
| **ATTracker** | **Ours** | Init. Info. (0.07%) Semi (40%) | NL+BBox | 40.6 **52.2** | 56.7 **69.6** | 40.9 **56.1** | 55.5 **67.6** | 77.5 **88.6** | 48.9 **60.4** | 47.5 **60.5** |

Despite only utilizing labels from the first frame in each video (0.07% of labeled data), our proposed method achieves competitive performance with fully-supervised baselines (i.e., SiamFC [3], TNLS-III [31], and TNL2K-2 [53]). Moreover, when the proportion of labeled data reaches 40%, the proposed ATTracker outperforms all other algorithms except JointNLT by a significant margin (at least 5% in SUC and 11.1% in PRE), irrespective of whether they are supervised or not. These results demonstrate the effectiveness of our approach for SS-VLT, establishing ATTracker as a strong baseline for future comparisons.

**OTB99.** The size of the OTB99 dataset is relatively small, with only 51 videos for training and 48 videos for testing. Compared with unsupervised visual trackers, our method ranks the top among them, surpassing LUDT by 7.4% in SUC and 11.7% in PRE. Meanwhile, the performance of our ATTracke is comparable to that of fully-supervised trackers such as TNL2K-2, SNLT, and JointNLT.

**LaSOT.** LaSOT provides a comprehensive benchmark for visual tracking, focusing on maintaining tracking accuracy over long sequences. With access to 40% labeled data, our semi-supervised tracker performs favorably against fully-supervised SOTA trackers including JointNLT, AutoMatch and SNLT. Notably, among unsupervised trackers, our ATTracker stands out as the top-performing method with only the label of the initial frame (0.07%), surpassing USOT by 15.2% and 15.0% in terms of the SUC and PRE.

**Qualitative Evaluation.** We conduct a visual comparison between the proposed ATTracker and several supervised trackers (such as JointNLT and SNLT), as well as the leading unsupervised method USOT, across four challenging sequences. These videos involve difficulties such as *vast appearance variations, view-point changing, semantic understanding, and occlusions*. As demonstrated in Fig.

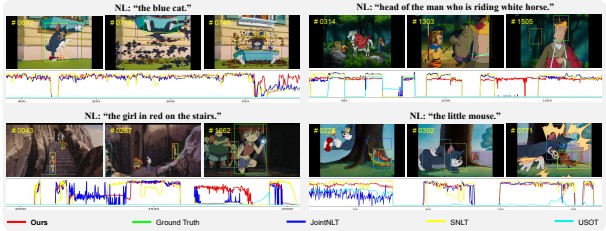

Figure 4: Qualitative evaluation of our proposed ATTracker on 4 challenging videos from TNL2K. Each sub-figure, arranged from top to bottom, is composed of the textual description, visualized localization results of key frames, and the IoU variation curves corresponding to different trackers. Best viewed in color and zoom in.

4, our approach performs competitively compared to supervised trackers, even with limited labels.

## 4.3 Ablation Study and Analysis

We first perform ablation studies to comprehensively analyze the impact of main phases within the consistencies-based training framework. Then, explorations of key designs within our approach for semi-supervised vision-language tracking are provided.

**Ablation of Main Phases.** To verify the contribution of each phase during training, we conduct ablation studies on six variants of our approach and fix the proportion of labels to 40%. The experimental results of these variants on all datasets are presented in Tab. 2.

- In **Setting** ①, we mainly rely on the pseudo label generated by the $X^2$-VLM for learning spatial consistency in

**Table 2: Ablation of main phases in the consistencies-based training framework.(%). SUC for Success and PRE for Precision.**

| Exp. Setup | Spatial | Temporal | Semantical | TNL2K | | | OTB99 | | LaSOT | |
|:---:|:---:|:---:|:---:|:---:|:---:|:---:|:---:|:---:|:---:|:---:|
| | | | | SUC | Norm.PRE | PRE | SUC | PRE | SUC | PRE |
| ① | √ | - | - | 39.4 | 49.1 | 32.3 | 51.7 | 68.8 | 45.9 | 34.7 |
| ② | - | √ | - | 36.5 | 39.6 | 25.4 | 48.5 | 57.5 | 38.2 | 21.0 |
| ③ | √ | - | √ | 43.6 | 57.0 | 41.7 | 55.8 | 76.6 | 52.3 | 47.7 |
| ④ | - | √ | √ | 38.5 | 48.4 | 29.0 | 50.7 | 59.7 | 42.5 | 29.1 |
| ⑤ | √ | √ | - | 45.7 | 61.9 | 47.0 | 62.1 | 83.6 | 54.5 | 54.1 |
| ⑥ | √ | √ | √ | **52.2** | **69.6** | **56.1** | **67.6** | **88.6** | **60.4** | **60.5** |

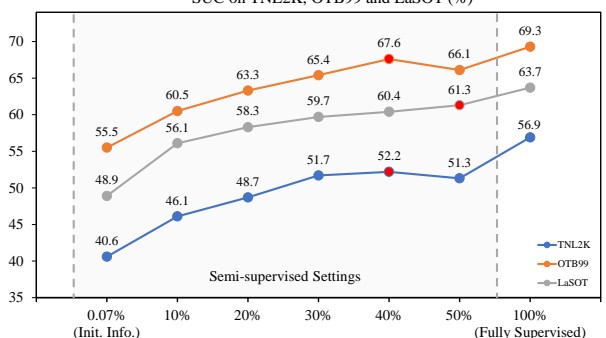

**Figure 5: The impact of the portion of ground truth labels used in our semi-supervised learning method (%).**

Phase I. In this variant, the ATTracker is trained via self-tracking within the same frame, aiming to acquire fundamental image-level perception ability.

- In contrast, **Setting** ② focuses solely on mining temporal consistency via bi-directional tracking in Phase II. The performance significantly degrades across all datasets compared to that of **Setting** ①, indicating that spatial modeling ability is more crucial for a transformer-based tracker.
- **Setting** ③ incorporates Phase III training to enhance semantical consistency of **Setting** ①. With proper optimization, the tracker can better perceive target from both modalities.
- Similarly, **Setting** ④ further utilizes the semantical consistency loss on the base of **Setting** ②, unleashing the power of VL representation learning for VLT.
- **Setting** ⑤ concentrates on utilizing supervision signals from both spatial and temporal aspects. However, without semantic guidance to redirect the tracker's attention, its performance is inferior compared to that of **Setting** ⑥.
- Our approach (**Setting** ⑥) leverages multi-consistencies from raw videos, achieving outstanding results compared to fully-supervised baselines and unsupervised methods.

**Impact of Portion of Labeled Data.** We follow the partition setting utilized in similar tasks [5, 49], where the proportion varies from 10% to 50%. Additionally, we introduce a novel partition setting for labeled data, considering how the VL tracker is initialized. In this setting, only the information from the initial frame is provided (0.07% of labeled data). As depicted in Fig. 5, with the increase

**Table 3: Ablation of key designs within our approach.(%).**

| Setting | TNL2K | | OTB99 | | LaSOT | |
|:---:|:---:|:---:|:---:|:---:|:---:|:---:|
| | SUC | PRE | SUC | PRE | SUC | PRE |
| $X^2$-VLM | 41.6 | 47.7 | 51.4 | 76.3 | 42.7 | 45.7 |
| plain | 46.7 | 48.2 | 63.3 | 84.2 | 55.6 | 55.7 |
| w/o. fine-tuned | 44.2 | 46.5 | 60.2 | 79.7 | 51.9 | 51.9 |
| **ours** | **52.2** | **56.1** | **67.6** | **88.6** | **60.4** | **60.5** |

of annotation proportion, the performance of the tracker gradually improves. When the proportion of labeled data reaches 40%, the tracker achieves its best performance, which is comparable to that under full supervision. Even with the initial information, our approach achieves competitive results compared with supervised base trackers and significantly surpassing the unsupervised methods.

**Impact of Pseudo Label Generation.** To ensure that the generated pseudo bbox in each frame contains meaningful content (i.e., the object highly related to the given language prompt), we fine-tune a large VLPM with the initial information in each video. The performance of the fine-tuned $X^2$-VLM shown in Tab. 3 indicates that it indeed possesses abundant VL knowledge that is beneficial for VLT task. Additionally, we employ the original $X^2$-VLM to generate pseudo labels in the **Setting without fine-tuning**, thereby evaluating the necessity of fine-tuning the VLPM.

**Impact of Designs in the ATTracker.** In the **Setting plain** outlined in Tab. 3, our asymmetrical attention layers within AT-Tracker are replaced with vanilla transformer layers, allowing other information to interrupt the template tokens. Consequently, this modification results in inferior performances across all datasets.

## 5 CONCLUSION

In this paper, we present a pioneering semi-supervised learning approach for vision-language tracking, which harnesses inherent consistencies in spatial, temporal, and semantical aspects of raw videos. We then propose the ATTracker, a simple yet effective model, to validate the effectiveness of our training pipeline. Extensive experiments confirm that the devised ATTracker serves as a solid baseline, outperforming both fully-supervised base trackers and unsupervised methods. By showcasing the potential of training a tracker with limited labels, we seek to draw interest and lay the groundwork for further exploration into semi-supervised vision-language tracking.

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

Received 20 February 2007; revised 12 March 2009; accepted 5 June 2009

