# OpenReview forum: "Consistencies are All You Need for Semi-supervised Vision-Language Tracking"
_acmmm.org/ACMMM/2024/Conference — MM2024 Poster_

### Official Review · Reviewer_8QJz · 2024-05-22

**Rating:** 5
**Confidence:** 3

**Summary:**

This paper employs various consistencies to achieve semi-supervised learning in V-L Tracking, particularly the introduction of semantics, which enables the method to surpass previous approaches, including supervised ones. The writing is clear and accessible, and the experiments are thorough.

**Strengths:**

1. Consistency is a crucial aspect of tracking. Identifying and leveraging these consistencies is vital. This paper is insightful in its use of multiple consistencies, effectively enhancing performance by correlating them.
2. The method leverages the semantic strengths of the V-L model to obtain more tracking results, which can inspire subsequent work.

**Limitations:**

1. The method adopts approaches from the BERT series instead of using newer LLM methods like Llama or CLIP. Appropriate comparisons or discussions are necessary.
2. There is a lack of discussion on the issue of consistency in semi-supervised related work. It is necessary to supplement the relevant literature.

**Suitability:**

3

---

### Official Review · Reviewer_8ahA · 2024-05-24

**Rating:** 4
**Confidence:** 3

**Summary:**

The paper presents a semi-supervised learning approach for Vision-Language Tracking (VLT). The authors argue that existing methods are overly reliant on extensive manual annotations, which are costly and time-consuming. To address this, they propose a novel scheme that leverages inherent consistencies in spatial, temporal, and semantic aspects of videos to generate high-quality pseudo labels and train the tracker with limited annotations. The method is validated using an Asymmetrical Transformer Tracker (ATTracker).

**Strengths:**

(1) The paper introduces a pioneering semi-supervised scheme that significantly reduces the dependency on labeled data.
(2) The paper includes extensive experiments that validate the effectiveness of the proposed framework.
(3) It provides a strong baseline for future research in semi-supervised vision-language tracking.

**Limitations:**

(1) The authors claim that their semi-supervised method outperforms fully supervised method, but they compare their approach under conditions of full supervision with other fully supervised approaches. And there are many fully supervised methods that are superior to the authors' method, such as UVLTrack(AAAI2024).
(2) On OTB99, the authors’ method enhanced with “Init. Info. (0.07%)” and natural language is not better than USOT(ICCV21) and LUDT(IJCV21).
(3) The authors have not conduct experiments to verify the effect of the data generation module on tracking performance.

**Suitability:**

3

---

### Official Review · Reviewer_mPyw · 2024-05-24

**Rating:** 3
**Confidence:** 3

**Summary:**

The paper proposes a semi-supervised learning approach for Vision-Language Tracking by leveraging multiple consistencies inherent in raw video data. In particular, the method introduces a progressive training framework on their Asymmetrical Transformer Tracker with three phases focusing on spatial, temporal, and semantic attributes of videos.

**Strengths:**

* The design on consistency learning seems reasonable and interesting.
* The authors provide detailed discussions in the related work about the development of Vision-Language Tracking, which offer a clear view of this field.

**Limitations:**

* Why combine static and dynamic as "template", but search area, language prompt, and special token [REG] as "others" in Asymmetrical Attention Module? Having more meaningful discussions and conducting detailed experiments about the separation of these could enhance the readers' understanding.
* The authors have cited [13], but what is the reason for not comparing to it? Since [13] discussed the asymmetrical design extensively, it is crucial to compare it with the Asymmetrical Attention Module in this work for ablation study.
  - [13] Divert More Attention to Vision-Language Tracking - NeurIPS 2022
* The comparisons to the recent baselines and the experiment on semi-supervised learning are insufficient. Since the main focus of this work is semi-supervised learning, but the comparing baselines are outdated - ICCV 2021 (not even using Natural Language as an additional source), which is not convincing enough to claim the benefits.
  - Also, here are some recent baselines on supervised learning that the authors are not comparing with:
    - Towards Unified Token Learning for Vision-Language Tracking - TCSVT 2023
      - TNL2K (SUC): $\textbf{58.6}$, OTB99 (SUC): 70.5, LaSOT (SUC): 70.0
    - All in One: Exploring Unified Vision-Language Tracking with Multi-Modal Alignment - MM 2023
      - TNL2K (SUC): 55.3, OTB99 (SUC): $\textbf{71.0}$, LaSOT (SUC): $\textbf{71.7}$
    - The performance in this work
      - TNL2K (SUC): 56.9, OTB99 (SUC): 69.3, LaSOT (SUC): 63.7

**Suitability:**

3

---

### Official Review · Reviewer_jtP1 · 2024-05-26

**Rating:** 3
**Confidence:** 3

**Summary:**

This work proposes to perform semi-supervised learning for vision-language tracking. Under the semi-supervised setting, only up to 50% of annotations are available during training. This work also proposes an Asymmetrical Transformer Tracker for the proposed semi-supervised task.

**Strengths:**

1. Introducing the semi-supervised visual-language tracking task.
2. The temporal consistency alignment is reasonable.

**Limitations:**

1. Comparing the semi-supervised setting with the unsupervised setting is unfair as shown in Tab.1, as the unsupervised setting doesn't provide any annotation except raw videos and the semi-supervised setting can utilize up to 50% BBoXs and language annotations. Comparing the ``init info'' setting with the unsupervised setting is also unfair, as ATTracker is empowered with a vison-language pretrained model giving clues of the object description and its visual appearance.
2. The ATTracker is claimed to harness inherent consistencies of raw videos while the consistencies are directly measured with the output of the data generated from the pseudo annotator fine-tuned on the vision-language pretrained model. The generated data are considered as pseudo-ground-truth, regarding less potential false positive annotations.


Minimal:
The red point highlights 50% of the annotations for LaSOT in Fig. 5.
Lacking \hat in Eq. 5 for the backward box.

**Suitability:**

3

---

### Meta-Review · Area_Chair_7vYK · 2024-06-30

**Recommendation:** Accept (Poster)
**Confidence:** 5

**Metareview:**

This paper was reviewed by four experts in the field. This work Introducing the semi-supervised visual-language tracking task, the proposed temporal consistency alignment is reasonable, the extensive experiments  well validate the effectiveness of the proposed method.

The reviewers raised concerns regarding experiments, comparison with previous methods, some details of the method, etc. After rebuttal and discussion period, all the reviewers gave positive ratings. The AC checked the rebuttal and was convinced that the authors well addressed this concern. The AC recommend this paper for acceptance.